# Family Needs Checklist: Development of a Mobile Application for Parents with Children to Assess the Risk for Child Maltreatment

**DOI:** 10.3390/ijerph19169810

**Published:** 2022-08-09

**Authors:** Heidi Rantanen, Irja Nieminen, Marja Kaunonen, Emmanuelle Jouet, Lidia Zabłocka-Żytka, Giovanni Viganò, Cristina Crocamo, Henrike Schecke, Giedre Zlatkute, Eija Paavilainen

**Affiliations:** 1Health Sciences Unit, Faculty of Social Sciences, Tampere University, Arvo Ylpön katu 34, 33520 Tampere, Finland; 2Pirkanmaa Hospital District, Tampere University Hospital, Elämänaukio 2, 33520 Tampere, Finland; 3School of Health Sciences, Tampere University of Applied Sciences, Kuntokatu 3, 33520 Tampere, Finland; 4Mental Health and Social Sciences Research Laboratory, Groupement Hospitalier Universitaire, Psychiatrie & Neurosciences (GHU-PARIS), 258 Rue Marcaret, Bât N, 2ème étage, 75018 Paris, France; 5Institute of Psychology, The Maria Grzegorzewska University, Szczęśliwicka 40, 02-353 Warszawa, Poland; 6Synergia s.r.l., Via Molino delle Armi 19, 20123 Milan, Italy; 7Department of Medicine and Surgery, University of Milano-Biocca, Via Cadore 48, 20900 Monza, Italy; 8Department of Addictive Behaviour and Addictive Medicine, LVR-Hospital Essen, University of Duisburg-Essen, Virchocstrasse 174, 45147 Essen, Germany; 9School of Medicine, University of St Andrews, N Haugh, St Andrews KY16 9TF, UK; 10Etelä-Pohjanmaa Hospital District, 60220 Seinäjoki, Finland

**Keywords:** child maltreatment, risk assessment, primary prevention, early identification, family, checklist, e-health, mobile health, mobile application, digital health application

## Abstract

Child maltreatment (CM) has been enormously studied. However, a preventive practice still requires comprehensive and effective instruments to assess the risks for CM in a family context. The aim of this study is to describe the development process of an evidence-based CM risk assessment instrument (Family Needs Checklist, FNC) for primary prevention online utilization. This article reports the development process of the checklist and its mobile application, consisting of a systematic literature review, identification of known risk factors using the content analysis method, and generation of the checklist, including a multidisciplinary group in the design and feedback. As a result, a comprehensive and compact checklist was developed to be used by parents or caregivers as a self-referral instrument with an option to be used with professionals as a basis for joint conversations. The FNC consists of parental, family-, and child-related risk factors. Based on the international evidence, the online application consists of knowledge about different CM types, information about risk factors and protective factors as well as recommendations and guidance to support services. The FNC is based on robust evidence on known risk factors causing CM in families. It can be used for primary prevention utilization in the general population.

## 1. Introduction

Child maltreatment (CM) is defined as physical, emotional, and sexual violence, including violent punishment and neglect of children aged under 18 years committed by parents or caregivers [1]. It also means living in a violent home environment and witnessing violence between parents or caregivers or parents and siblings [2,3,4] or being a victim of parental alienation [5]. Violence is defined as the intentional use of physical force or power threatened or actual resulting in injury, death, psychological harm, maldevelopment, or deprivation and is, therefore, more than acts leading to physical harm [1]. As opposed to violence, child maltreatment may result from an unintended action by the parent caused by a lack of health, knowledge, ability, or resources, a failure to act, or carelessness that causes actual or potential harm to the child’s health, coping, development and/or dignity [6]. According to Krug et al. (2002), violence is distinguished from unintended events that result in injuries. Intention to use force does not automatically mean an intention to cause damage. In fact, between intended behavior and intended consequence, there may be a considerable disparity. One may intentionally commit an act that, by objective standards, is judged to be dangerous and likely to result in adverse health effects but may not perceive it as such [7]. According to Henschel et al. (2014), self-control is seen as an individual’s relatively stable capacity to control emotional expression and behavior, but under stressful circumstances, it can be depleted [8]. Indeed, parents may experience exhaustion [9], fatigue, haste, loss of temper, distraction, dissatisfaction, stress, or weakness that triggers maltreatment of their child [10]. Parents may unintentionally repeat harmful parenting practices experienced in their childhood and pass on maltreatment to the next generation [11,12,13,14,15,16]. Parents may also live in social isolation and lack knowledge about the normal development of the child and positive childrearing practices [17]. In parenting, unintentionality is used, for example as unintentional neglect [18] or unintentional injuries [19]. Intended or unintended violence against children has lifelong and devastating consequences for a child’s physical, mental and social health [1,20] and is therefore criminalized in many countries [21,22].

Globally, almost 300 million children aged 2–4 experience physical harsh discipline, and/or psychological violence within their families [23]. Children also face the highest risk of homicide by parents [24]. In Europe, the total estimation for any acts of violence, moderate or severe violence against children is over 80 million [25]. There is extensive research around the world documenting the epidemiology, consequences, patterns, or syndromes of maltreatment as well as risk and protective factors [26]. Krug et al. (2002) have stated that greater priority should be given to the primary prevention of violence—that is, instruments to stop CM from occurring in the first place. Different sectors and agencies should be engaged in prevention activities, and evaluation should be an integral part of all programs [7]. Subsequently, the WHO has launched INSPIRE, seven strategies for ending violence against children [1]. The WHO Regional Committee for Europe published an action plan for 2015–2020 to prevent child maltreatment in Europe. The purpose was to reduce the prevalence of child maltreatment by implementing preventive programs that address risk and protective factors, including social determinants [27,28]. Many European countries, for example, Finland, have followed this action plan [29]. The ERICA project Stopping Child Maltreatment through Pan-European Multiprofessional Training Programme: Early Child Protection Work with Families at Risk (2019–2021) invested to prevent and combat child maltreatment, by building the expertise of professionals concerning minors living in families with child maltreatment risk. In the ERICA project, an evidence-based risk assessment instrument, Family Needs Checklist, was developed for the early prevention of child maltreatment [30,31]. To support parenting, several interventions with strong evidence have been developed to reduce child behavioral problems, such as the Incredible Years parenting training [32,33,34] and Strongest Families [35,36]. In Finland, early and preventive services operate to an increasing extent collaboratively at all levels of CM prevention. In particular, many low threshold services offer professional education and support for parents in collaboration with child welfare services providers, for example, the Federation of Mother and Child Homes and Shelters [37], Miessakit Association [38], Maria Akatemia [39], MIELI mental Health Finland [40] and AddictionLink [41]. They also provide low threshold services as an independent operator.

To stop CM from happening in the first place in families, we need to consider several aspects. First, there is a need to consider all families with children because CM and family violence exist across all strata of society [42]. In the communities, CM prevention is usually practiced on three levels. Primary prevention targets all parents regardless of risk for CM. It focuses on reinforcing beliefs, practices, and conditions in the community and culture. Secondary prevention programs target parents that are more at risk of CM, whereas tertiary prevention targets parents after occurred CM and attempt to prevent the reoccurrence of CM [42,43]. CM prevention has been less focused on the primary prevention level [7]. From the public opinion perspective, the most important things to prevent CM are education (what CM is and how to raise a child), raising of awareness, reduction in substance abuse and poverty, and access to services and support [44]. According to Devries et al. (2018), parents may be more likely to report less severe acts of violence than more severe forms of maltreatment. For example, the emotional maltreatment of children appears to be more common than physical maltreatment and may not be viewed as particularly traumatic [45]. This indicates the need for parental education on various maltreatment types. Risk-informed policies and programs across all sectors as protective factors [46] support public awareness-raising through the education of parents and children [42]. From the professional’s perspective, the child and family service professionals should hold expertise in the use of CM knowledge, inter-agency collaboration, support services [6,47], and risk assessment instruments [6].

Second, known risk factors as well as protective factors play an important part in the identification of familial risk conditions for CM. Therefore, they must be reflected [46]. In Belsky’s (1993) developmental–ecological model, risk factors for child physical maltreatment are organized around three conceptual contexts: the developmental, immediate interactional and broader context. The developmental context considers parental and child characteristics and processes, including the intergenerational transition of CM. The immediate interactional context considers parenting and parent-child interaction. The broader context considers communal, cultural and evolutional risk factors [48]. Familial CM risk conditions can constitute single or multiple risk factors. It is stated that no single risk factor is necessarily indicative of CM, but the cumulation of the risk factors increases the risk for CM [6,43]. According to Belsky (1993), risk factors do not unequivocally cause parenting disturbances. It merely occurs when stressors outweigh support and risks become greater than protective factors [48]. Therefore, the unique situation of the child and family must be considered as a whole to plan and offer support services based on family needs [6]. There are differences in how the risk factors are highlighted in the guidelines or strategies. They are also known to vary between cultures [1]. For example, the guideline set by the National Institute of Health and Care Excellence, UK (NICE) [49] focuses on (1) how to recognize and respond to child abuse and (2) neglect and when to suspect the maltreatment of children under 18 in the UK. As stated, the guideline is based on well-recognized risk factors, although they are not presented to the reader [49]. The guidelines set by the Centers for Disease Control and Prevention, USA (CDC) [50] and WHO INSPIRE strategy [1] focus on the prevention of CM and include individual, relational, community, and societal risk factors. The Finnish Clinical Practice Guideline [6] focuses on CM prevention in the family. It includes parent, family situation and child-related evidence-based risk factors and identification instruments of the CM. The guideline was originally published in 2008 [51] and updated in 2015 [52] and 2022 [6] (The latest English shortened version is available online: https://www.hotus.fi/wp-content/uploads/2022/06/hoitosuositus-lasten-kaltoinkohtelu-eng-web.pdf (accessed on 22 July 2022)).

Third, extensive research is available regarding the CM identification instruments, [43,51,52,53,54,55,56]. The majority of the instruments are developed for clinical use for detecting the potential, substantiated, or recurrence of different types of CM [43,53,54,55,56,57,58,59,60] or its prevalence and depth, such as ICAST [61,62,63]. Many instruments are reported to have challenges with content validity, internal consistency, reliability, measurement error, structural validity, hypothesis testing, cross-cultural validity, or criterion validity [55,56]. Additionally, the risk assessment instruments are usually developed for use in the secondary and tertiary levels of prevention [43] and not for the primary prevention purposes. In accordance with Winters et al., checklists can be utilized to assess the family situation because they have powerful potential to improve family outcomes. Checklists democratize knowledge and help ensure that all parents receive evidence-based knowledge to guide the evaluation process. A checklist standardizes the process to ensure that all elements are addressed and the same information is available to parents as well as professionals to facilitate conversations. They also have the potential to reduce the costs of health and social care [64].

Fourth, child maltreatment or its risk conditions in the family are extremely sensitive issues and at the same time need urgent attention. Child maltreatment or its risks can be identified by a professional, parent, caregiver, a child or adolescent [65]. Parent and child self-reports are shown to be an important part of the evaluation process. A recent review revealed that parent or child self-reports increase CM prevalence, compared to reports made solely by professionals. Self-reports make CM more visible [65]. Besides self-reporting, parents need instruments that provide information on CM and opportunities to reflect their own life situation in light of the CM phenomenon. On the secondary and tertiary prevention levels, risk families are usually screened in the clinical contexts [43]. Primary prevention level includes universal services that are equitable, cost effective, and non-stigmatizing by nature. They should be available for anyone and can be designed to respond to several problems [42]. When screened by professionals, parents may experience fear of being caught and stigmatized when evaluated as objects. They may respond to inquiries in a socially acceptable way or distort the true state of conditions in the family [42]. For example, recent reviews found biomarkers to be probably a more reliable method for identifying substance use in pregnant women than self-reporting [66,67]. Research on CM has relied less on the anonymity of the respondents, and anonymous responses are considered the only viable way to elicit an honest response [68]. On the primary prevention level, universal services should allow self-referring and easy access to services that meet their preferences and needs. The prevention of CM inquires holistic approaches through collaborative partnerships [42]. Self-evaluation is the self-referential feedback process by which we survey the course and actions of our success. It allows us to intervene and adjust our behavior [69] (p. 254). Self-evaluation of CM or its risk conditions in the family requires research-based knowledge on (1) what are harmful conditions, (2) what causes harmful conditions, (3) why it is harmful to the child, (4) how and where to access support. The success of mobile applications has increased in the past years. Parents have requested web-based ways to describe and express their worries to professionals [70].

CM is widely studied but research has not been sufficiently utilized in early identification and prevention practice. In the ERICA project, the focus was on evidence-based risk assessment practices in the multicultural context, the development of a risk assessment instrument for multi-professional purposes, and an online risk assessment application creation for parental use. The purpose of this development study is to reduce CM occurrence with a risk assessment instrument, the Family Needs Checklist (FNC) and its mobile application. The user manual of the checklist for training and multi-professional use provides professionals evidence-based knowledge about CM to enhance joint conversations with parents. The mobile application of the FNC is intended for parents. It increases parents’ knowledge about CM, helps a parent reflect on their unique situation regarding evidence-based CM risks and protective factors. It also enhances parents’ ability to seek low threshold or advanced professional support. There is a clear need for a family CM risk assessment instrument that (1) is comprehensive and can be used by all parents and caregivers at any life point, (2) can be used at the primary prevention level as a self-referral instrument for parents or caregivers, (3) includes knowledge about different maltreatment types, risk factors with explanations and, protective factors, and (4) has the potential for various child and family service professionals’ supportive online or in-person encounters in the process of reducing CM. The aim of this study is to describe the development of an evidence-based CM risk assessment instrument for primary prevention online utilization. The development process includes a systematic literature review of the risk factors for CM in the family context analyzed with the content analysis method followed by the creation and development of the evidence-based checklist and its mobile application.

## 2. Materials and Methods

In the development of the Family Needs Checklist (FNC), an approach for checklist development established by Winters et al. [64] was utilized. It is a three-phase development process, including (1) a review of the existing literature, (2) understanding the needs and workplaces of the users, including a multidisciplinary group in the design, and (3) using an iterative approach for the rigorous pilot testing and validation of the checklist [64]. In the present study, phase one and partially phase two, including a multidisciplinary group in the design, were completed. Phase one involved compiling and updating the evidence base for content and the creation of a preliminary checklist. A systematic review of databases was completed to retrieve review articles focusing on risk factors of CM in families with children and risk assessment tools. The review was part of the update process of the Finnish Clinical Practice Guideline that was completed in 2022 [6]. For the systematic literature review update, the research questions were as follows: (1) What are the factors related to the risk conditions of child maltreatment? (2) What tools are reliable in identifying risk conditions in the family? We relied on international evidence summaries that strengthen the evidence base. Additionally, we gathered evidence that answered questions on why, for example, young parenthood is potentially a risk factor and how it can be harmful to the child, to be able to build a feedback section of the FNC online application. The JBI (Joanna Briggs Institute) guidelines for systematic reviews [71] were conducted in cooperation with the Finnish Nursing Research Foundation experts [72]. The foundation hosts The Finnish Centre for Evidence Based Healthcare: A JBI Centre of Excellence [72]. The following databases were searched: Cochrane, DARE, JBI, PROSPERO, PubMed/ Medline Ovid, CINAHL, Academic Search Ultimate, PsycINFO, SCOPUS, and MEDIC. The search included both MeSH search terms and keyword synonyms, using a search strategy developed by a librarian specialist in Medline and subsequently modified for use in the remaining databases. References in related review articles were also reviewed. Reviews were included in the literature review if they addressed either the risk of child maltreatment or the method used to assess it. Searches were confined to systematic reviews, scoping reviews, meta-analyses and -syntheses published in English, Finnish or Swedish between 1 January 2014 and 30 April 2021. We excluded single studies and systematic reviews whose focus deviated from child maltreatment in the family or its risk conditions. The quality of evidence was examined by two reviewers using the JBI Critical Appraisal Tool for Use in JBI Systematic Reviews (https://jbi.global/sites/default/files/2019-05/JBI_Critical_Appraisal-Checklist_for_Systematic_Reviews2017_0.pdf, accessed on 18 July 2022). The data were analyzed using the content analysis method. The PICO, search strategy and flowchart figure of the systematic review and meta-analysis selection process are available as (Appendix A: Research questions, PICO, search strategy and flowchart of the systematic review and meta-analysis selection process).

The identification and conceptualization of the CM familial risk factors were based on the Finnish National Nursing Guideline (2015) [52] and reinforced with the research data from updated guideline [6]. The evidence base of the FNC was compared with knowledge from the CDC guideline [50] and the WHO INSPIRE strategy [1]. During the ERICA project, we were able to use also data from Scotland [73,74,75] and Finland [29] regarding risk factors. We included them in the comparison process to validate and strengthen the findings. The statements for the FNC were generated from the found risk factors, for example, the risk factor “young parent” produced a statement: “My age is ≤18”. The initial Family Needs Checklist consisted of 55 statements, 5 demographic risk factors and 3 risk factors for shaken baby syndrome.

Phase two involved receiving feedback on the wording and design from the trainees and experts who attended the ERICA project. Further details are available elsewhere [31,76]. The checklist was reviewed by trainees and/or experts in the ERICA training cohorts 1 and 2 that were held between December 2020 and May 2021. The FNC was included in one of the training modules and introduced and discussed during the training session. The feedback was given during the ERICA training cohort 1 and cohort 2 in six European countries (the UK, Finland, Germany, Italy, Poland, and France). The feedback was gathered by trainers and delivered via e-mail to the development team. The checklist was modified and finalized based on feedback for the online application. The development process is presented as (Appendix A: Outline of the modified checklist development process used in developing the Family Needs Checklist). The technical development process of the mobile application was implemented by two data professionals at Tampere University of Applied Sciences and made available in the Google Play Store. Where applicable, the Qvalidi 2019 [77] checklist was utilized in the FNC development process to help identify early-stage issues which needed to be considered in application development to enhance the application’s suitability for users and to strengthen its validity and reliability. The Qvalidi 2019 checklist includes 49 items and four sections, namely health-related content, technical features, user experience, and safety. It provides a comprehensive view of the strengths and potential areas of improvement of digital health and wellness applications [77]. The completed Qvalidi 2019 checklist is available as (Appendix A: The report of the Family Needs Checklist (FNC) online application using the Qvalidi 2019 checklist).

## 3. Results

A total of 42 articles answered our research questions; 32 articles were included in the checklist development: combined meta-analyses and systematic reviews (*n* = 12), meta-analyses (*n* = 2), systematic reviews (*n* = 16), scoping review (*n* = 1), and integrative review (*n* = 1) consisting of a total of *n* =1660 studies. A table outlining included articles in the checklist development is presented in (Appendix A: Reviews included in the evidence extraction process). The 10 articles that answered the second research question were part of the ground work of the development process and are therefore utilized in the introductory part of this article. We did not find any preventive instrument suitable for our purpose. Therefore, we developed a new instrument: the Family Needs Checklist (FNC) and its mobile application.

The FNC is formulated to increase CM knowledge for both parents and professionals and enhance collaborative encounters. The FNC mobile application is meant primarily for the parents’ or caregiver’s self-referral use. The mobile application consists of six sections: (1) an instruction page for users, (2) an information section about different types of child maltreatment, a statements section with binary yes/no options concerning (3) the parent or caregiver risk, (4) the family life and situation, (5) the child risk factors, and (6) feedback section including causes and consequences of the CM risks, protective factors protecting from CM, and encouragement to seek support as per need, as well as low threshold support service information and www-links. The user manual of the FNC can be used by professionals as a source of evidence-based CM knowledge or in training purposes. It enhances collaborative encounters with the parents. The manual can be found on the homepage of the ERICA project [30]. In the FNC, the risk factors are incorporated either in the statements, feedback, or information section. For example, the risk factor “*Poor parenting practices*” is incorporated in the feedback because it is seen as one consequence of experienced maltreatment in the parents’ childhood [3,11,12,13,78]. In the formulation of the FNC, we aimed to make a distinction between the cause and the consequence. The causes form the statements, and the consequences form the feedback section.

### 3.1. Factors Related to the Risk Conditions of Child Maltreatment

As a result of the literature review synthesis, risk factors concerning the family context were constructed into three categories: risk factors concerning the parent or caregiver, the family situation, and the child [6]. Categories and contents are presented in Table 1.

Some of the countries were able to provide evidence about nationally discovered CM risk factors. Therefore, we compared the checklist with the risk factors discovered in the WHO INSPIRE strategy [1], and CDC guideline [50] as well as risk factors identified in Scotland [73,74,75] and Finland [29]. The comparison is presented in Table 2.

The comparison shows that the FNC includes all risk factors but not straightforwardly sex, associating with delinquent peers, nonbiological parents, forced marriages, chaotic lifestyles of the parent, inadequate housing, lack of emotional bonding and poor parenting practices.

Regarding the risk factor “*sex*”, there is robust evidence that girls have a two-fold higher risk of becoming sexual abuse victims compared to boys [79]. Additionally, there is increasing evidence regarding female perpetration of the violence [80,81]. However, the FNC as a primary prevention self-referral instrument must consider all the potential sexes. Even sexual violence against the child is considered from the unisex perspective in the FNC information section. In the FNC, the risk factor “*delinquency*” is placed under the sub-category of history of antisocial behavior and criminal history. According to our literature review, we did not find evidence about “*nonbiological parent*” as a risk factor. Instead, there is evidence that the absence of a biological parent increases the neglect of a child [23]. Regarding the risk factor “*lack of emotional bonding*” we found that insecure adult attachment increases child maltreatment risk by increasing emotion regulation problems, minimization of emotions, and distancing oneself from others [14]. These are included in the feedback section of the FNC. In the FNC, “*poor parenting practices*” or “*parents’ violent behavior*” is found to be a result of experienced maltreatment in childhood increasing negative parenting practices [3,13]. Therefore, the factor was included in the feedback section. Additionally, information about physical violence is in the CM info section of the FNC. Risk factors namely “*forced marriage*, *parents’ chaotic lifestyles or inadequate housing*” are not found in the FNC as such. They are either culture specific or might refer to poverty, such as a cramped home, mental illness, stress, substance abuse or domestic violence, which were found in the systematic review. These risk factors may require further exploration.

The FNC covers even more CM risk factors than presented in the guidelines under comparison. All comparisons included risk factors, namely maltreatment experienced by parents in their childhood and prior trauma, physical and mental health problems, single or divorced parenthood, intimate partner violence, family violence and problems in close relationships and child with a disability. Risk factors found only in FNC are a child with learning problems and an excessively crying infant.

### 3.2. Feedback from the Trainees and Experts

One of the main results of this study is that experts and trainees from different countries had different opinions and feedback on the FNC giving comprehension of the specific cultural dimensions of each country. According to the feedback, some of the referees felt that the FNC is too intrusive, that is, it deals with topics that are too intimate and that can make both parents and professionals feel uncomfortable. For example, the statement about the criminal history was seen as very strong. Some of the statements were considered especially important risk factors. Statement about the parent’s history of childhood maltreatment was especially seen to require available professional support services linked to the statement. The FNC was also seen as potentially challenging to parents. This could prevent parents from being completely transparent. Some referees considered statements also a lot to think about, good stimuli for discussion, and help start a conversation between parent and professional. Some questions were seen as stigmatizing. Some of the referees did not see themselves using the FNC as part of a family interview to assess the risk of maltreatment, but they did find it a good basis for discussion among professionals about the local understanding of maltreatment. All feedback and modifications are available as Appendix A (Appendix A: Initial Family Needs Checklist (FNC) mobile application, feedback summary, and modifications to the final version).

### 3.3. The Content of the Refined Family Needs Checklist

As a result of the literature review synthesis, the FNC reflects risk factors concerning the family context and concludes with three perspectives: risk factors concerning the parent, the family situation, and the child [6]. Fifteen statements concern the parent or caregiver risk factors, seven statements concern the family life and situation risk factors, and six statements concern the child risk factors.

Statements concerning the parent, caregiver, or adult
^1.^ I was maltreated as a child [11,12,13,14,15,16,17,78,82,83,84].^2.^ I have experienced traumatic events as a child and have not gotten over them [16,84,85].^3.^ I sometimes have unrealistic expectations for what the child’s behavior should be like [9,12,17].^4.^ I am not always able to control my child’s disobedient behavior [14,16,17,86].^5.^ My age is ≤18 [12,13,14,16,17,87].^6.^ I have no formal education or a low level of education [12,85].^7.^ My life is currently stressful [14,17,85,88,89].^8.^ I use substances such as tobacco, drugs, and/or alcohol [16,17,88,90,91,92].^9.^ I do not know what the safe limit for alcohol use is in a family with children [17,85,92].^10.^ I have a history of antisocial behavior or criminal offenses [12,16,88].^11.^ I am suffering from mental health problems, such as depression or a feeling of worthlessness [12,13,16,17,84,87,88,93,94,95].^12.^ I have to use a lot of health services with my child because my child is often sick or unwell [95].^13.^ I experience difficulties in taking care of my child’s basic needs, for example, dental hygiene, basic hygiene, clothing, or healthy food [12,14,96,97,98].^14.^ I experience difficulties in taking care of my child’s basic needs, for example providing my child with social and emotional support [12,13,17,85,99].^15.^ I experience difficulties in taking care of my child’s basic needs, for example, schooling and sleeping times [12,13].

Statements concerning the family life and situation:
^16.^ I am a single or divorced parent [12,85].^17.^ I have three or more children [12,85].^18.^ There are constant financial worries or unemployment in my family [12,16,17,85].^19.^ My child/children are three years old or younger [9].^20.^ I feel lonely and have not had enough support from my community, relatives, friends, or spouse [17,100].^21.^ I have experienced intimate partner violence at home [3,12,14,15,16,17,85,93].^22.^ My child has experienced intimate partner violence or other form of family violence at home [3,86,101].

Statements concerning the child:
^23.^ My child had complications associated with pregnancy or birth [12].^24.^ My child cries a lot [9].^25.^ My child has been diagnosed with a developmental or physical illness or has challenges related to emotions or social situations [89,100,102,103].^26.^ My child is often disobedient, misbehaving, difficult, or irritable [89,102].^27.^ My child has challenges at school or day care [95,99,102].^28.^ I have to calm my child down by giving him/her drugs, sedatives, or other substances [95].

### 3.4. The Content of the Family Needs Checklist Mobile Application

The FNC is embedded in a mobile application. The instruction page consists of a short explanation and the purpose of the application. The application is intended for use by expectant parents, parents, or other persons caring for children or any adults who want to think about the safety and welfare of their own family. It is highlighted that all families will meet many challenges in their lives. The challenges can be positive or negative, anticipated or unprecedented, and may pass quickly or be recurring. These challenges may develop into insuperable problems, especially when they accumulate. The purpose of the application is to help the parent to reflect on their present situation or that of their children, grandchildren, or other children they take care of and of the whole family, and to identify any possible needs for a change or early support. The parents are informed that the statements are based on research knowledge and cover a wide range of safety-related challenges in the child’s family life. The parent is advised to consider whether or not each statement applies to their situation. It is advised, too, to be fairly honest with themselves when answering the questions, because, at the end of the statement section, they will receive feedback based on answers, which includes research-based information on risk factors, protective factors, and available online or via phone services. The privacy policy information is available for users through the instruction page.

The information section contains basic information about various maltreatment types. It answers the questions about what child maltreatment, physical violence, emotional violence, sexual violence, neglect, intimate partner violence, or domestic violence are all about, and what consequences can there be for shaking a baby. Going through this section is voluntary. It is meant to be informative only. The statements section, with binary yes/no options, consists of evidence-based known CM risk factors as presented above. After completing the checklist, the feedback section includes general positive feedback if there are no risk factors found. Concerning the risk factors found, the parent or caregiver will get research-based information about each CM risk and protective factors as well as contact information and online links or phone numbers to support services.

As a result of the development process, the FNC mobile application is available for free for validation in different cultures. The Family Needs Checklist mobile application is available in the Google Play store in German, Italian, Finnish, and English languages. Currently, the Finnish and English versions include www-links to support services in Finland only. It is highlighted that the checklist is under the validation process in Finland.

## 4. Discussion

The purpose of this study was to describe the development of an evidence-based CM risk assessment instrument for primary prevention online utilization. Our study addressed the need for a family CM risk assessment instrument that (1) is comprehensive and can be used by all parents and caregivers at any life point, (2) can be used at the primary prevention level as a self-referral instrument for parents or caregivers, (3) includes knowledge about different maltreatment types, risk factors with explanations and, protective factors, and (4) has the potential for various child and family service professionals’ supportive online or in-person encounters in the assessment process.

Evidence-based and validated instruments help identify parental worries and start a collaboration with the family to find out more about the family’s situational possible support needs [6,51,52]. Instruments used for CM prevention need to be broad enough and take into account issues that are considered risks and express parental worries [6]. The systematic review of the known risk factors for CM revealed the latest evidence and serves as the basis for the FNC and its online application. Compared to few other guidelines [1,29,50,73,74,75] it is found to cover well the phenomena of CM risks and risk conditions. Risk factors found only in FNC are “*a child with learning problems”* and “*an excessively crying infant*”. According to the systematic reviews, the child may have learning problems due to medical condition, tiredness, problems in concentration, experience of bullying at school, poor relationships at home or school, lack of friends as well as problems in the home environment, such as lack of emotional support, neglect from daily guidance and nurture, witnessing domestic violence or experiencing physical, emotional or sexual violence [95,99,102,104]. An excessively crying infant may have a challenging temperament or have stomach symptoms, such as flatulence. A parent may find the baby irritable, overactive, or otherwise difficult to handle. Parents may have unrealistic expectations for the baby’s ability to control his or her behavior. They may feel hopeless when they are unable to cope with a challenging situation [9,105]. When a baby is constantly crying, the family may experience despair and a sense that their everyday life is in shatters. Excessive crying can also interfere with breastfeeding, isolate parents, strain and break family relationships, cause feelings of parental failure or lead to physical and mental exhaustion. Such a situation may put the baby at risk of child maltreatment, increase problems later in life or even death [9]. Parents discovering such risk factors via the FNC may become encouraged to seek professional help early instead of trying to manage issues themselves at home. Additionally, the FNC covers well the causes and consequences of the CM to the child’s welfare as well as known factors that protect children [3,9,11,16,17,46,85,86,89,106] from CM in different circumstances. The checklist can be used by any parent or caregiver at any time, although the online application requires online access. In a future, we are proceeding to make the FNC also available in paper and pencil format. This is especially advantageous in the joint conversations between parent and professional. This FNC mobile application is unique in the sense that parents and caregivers can access it online and fill it out on their initiative when they are concerned about their family’s situation and then seek support based on it. Parents may also be asked to complete the FNC mobile application prior to the appointment, where any issues the parent has identified can be discussed. There are no other similar checklists.

From the prevention perspective, screening for CM before the maltreatment occurs promotes the early identification of CM risks, which is necessary to swiftly refer children and their families to early intervention programs and support services [6,7,54]. By acting preventively, many of the parent-, family-situation- or child-related CM risk issues can be managed also on a low threshold level of the community or even without any professional support. The FNC is developed to serve parents as a self-referral instrument to early recognize and prevent CM. In primary prevention, evidence-based assessment instruments should be used systematically in all child and family services. The FNC can be introduced to parents in any communal parent–child society. The evidence-based knowledge of the FNC can be delivered by professionals in parental groups, for example, in daycare, and schools. The FNC can be used also in adult health, and social and mental services during joint discussions about child welfare in the family. Certain legal restrictions must be adhered to, for example, restrictions on asking about the parents’ or caregivers’ criminal records which can be highly classified information. The use of the FNC should be based primarily on the parent’s assessment and need to discuss any concerns they may have identified. It also has a great value to inform professionals and students about CM risks, their causes, and consequences as well as protective factors and available support services. Reflecting on the risks of CM would be important to everyone, as the purpose of primary prevention is to raise awareness in society and educate the public to promote wellbeing.

The FNC aims to increase parents’ or any adult respondents’ knowledge about child maltreatment, encourage them to reflect on their situation, and serve as a channel for early support services and help. The FNC includes knowledge about different maltreatment types, risk factors with explanations of the causes and consequences as well as factors that protects CM from happening. The evidence-based knowledge is available both to parents and professionals. Mutual knowledge enhances the conversations and understanding the family situation as a whole. Therefore, the FNC has the potential for various child and family service professionals’ supportive online or in-person encounters in the assessment process. In a review that gathered parents’ and children’s experiences of child protection, parents emphasized the importance of transparency and regularity when working with families. Honesty and openness facilitate the building of trust in a collaborative relationship, which in turn contribute to positive results. Professionals should have time to listen to the feelings and views of family members. The availability of the same professional and the accurate answering of the calls are also seen as important [107].

Risk factors may vary across cultures [23]. That can make identification challenging. CM risks are many, and their emphasis varies from culture to culture, but they seem to have a similar base. In this study, experts and trainees from different countries had different opinions and feedback on the FNC, giving comprehension of the specific cultural dimensions of each country. The result has to be taken into consideration. It is recognized that cultural differences exist between countries and continents regarding which parental acts are regarded as abusive or neglectful [63]. This issue appeared while the content of the instrument was developed [76]. In addition, the differences were seen, for example, in the cultural ways to express the factual content of the statements which varied from forthright to a softer style of speaking. For example, in France, it should be remembered that it was not until 2021 that a National Consensus Process for a shared vocabulary of abuse of people in vulnerable situations was launched. In 2022, this definition was accepted in the recently passed Child Protection Policy Renewal Act (Law no. 2022-140 of 7 February 2022). The French professionals are not used to asking such direct questions to families. They are also reluctant to ask any questions that could appear stigmatizing, such as those regarding culture and ethnicity, religion, or origins, for those questions are not allowed to be asked in France. The mentioned law and the consensus should be part of a cultural change, such as the major pending case and debate that are currently in progress on the predocrimanility in the catholic church or the lay field of education [108]. In the cultural frame of reference, the focus seems to be on the ways the risks are discussed, local support services are arranged that fulfill the needs of the parents, and how the CM is prohibited by the law. Fortier et al. (2020) have studied what type of survey questions are identified by adults as upsetting in a focus of CM. The results showed that only 4% (*n* = 1000) identified maltreatment-related questions as upsetting [109].

Child maltreatment is a global, public health, human rights, moral and social problem [1,23,25]. At the same time, CM in the family is an extremely sensitive issue and requires a high standard of ethical methods in encounters with the families. The statements and feedback regarding risk factors may cause parental anxiety or self-inflicted trauma. By reflecting on own life situation through risk factors, CM consequences and preventive factors, the parent or caregiver may face emotions related to, for example, harmful parenting practices, experienced trauma, or adverse childhood experiences. The checklist also includes hard facts: on the instruction page and in the feedback section, it also includes a solution-based focus that problems belong to life and can be solved. Problems sometimes come as a surprise, or the family maintains them in their interaction system. If the family is unable to solve the problems on their own, they can be helped by support services [110]. The checklist also encourages parents to be courageous and act early since challenges in the family may develop into insuperable problems, especially when they accumulate. The family may lack factors that would protect their safety and welfare, such as the support of society or the close relatives, and ability to manage a difficult situation or a lack of information about how to act in the best way in a difficult situation. The checklist recognizes also that children are the most vulnerable to these situations and experience symptoms when they feel insecure. Most importantly, the checklist provides causes for the CM risk factors that are rooted in life experiences, poor development, lack of knowledge, resources, or support, and are not seen as inherent in the parent. A parent must acknowledge that there is a reason for the harmful behavior, other than what the parent is inherently, and that behavior can be changed through their willpower and resilience or together with professionals. The FNC online application encourages to seek early support if a parent feels the need through reflection and www-links to various support services, even in urgent situations.

The use of the FNC online application requires adequate support services and their rapid availability. The ethical aspects of the FNC online application are considered from the perspectives of laws as well as human rights conventions. For example, Finland has ratified the convention on children’s rights [111]. According to Finnish national laws, parents hold the legal responsibility for the child’s safety and healthy development [112,113,114]. Similarly, they have the right to receive adequate support to fulfill this responsibility. Social and health care service professionals hold the responsibility for the fulfillment of these rights together with the low threshold service providers [112,113,114]. This demands effective and creative multi-professional collaboration. For example, in Finland, the family centers have reinforced greatly interprofessional collaboration and collaboration with low threshold service providers by developing family centers [115]. CM is a multidimensional concept and comprises research knowledge from various disciplines. Inter-Organizational disconnection and lack of adequate collaboration between agencies is an acknowledged issue that requires further development [42,116]. The FNC and its manual for professionals, on its part, aim to unify inter-organizational connections and optimize the support service structure by delivering consistent evidence-based information on CM.

Future research directions include the validation of the Family Needs Checklist in the Finnish culture. The next steps are to understand the needs and workplaces of the checklist users including experts of experience, and social and healthcare professionals, and to use an iterative approach for rigorous pilot testing and validation of the checklist as well as develop an effective implementation strategy.

We would like to acknowledge a few limitations. First, the feedback was gathered without a systematic questionnaire and consisted of various perspectives. Therefore, we were neither unable to determine the actual number of respondents nor could we compare the feedback given by experts and trainees. Nevertheless, we were able to consider all the feedback effectively. Second, although the FNC application is at present an evidence-based instrument, it requires validation. Third, we were not able to compare the risk factors more widely within the European Countries. This may be accurate in future research projects.

## 5. Conclusions

The FNC is based on robust evidence on known risk factors causing CM in families. The Family Needs Checklist is the first one that is based on international, multidisciplinary, systematically reviewed robust research evidence of CM risks. Definitions and actions of CM vary between cultures and can make the identification of CM risks and risk conditions challenging. Based on multidisciplinary and international research evidence, the FNC is useful in different cultural contexts. It can be used for primary prevention utilization in the general population. It brings forward parents’ issues, which can be discussed individually, according to their needs. The use of the FNC and its online application requires adequate support services and rapid availability.

## Figures and Tables

**Table 1 ijerph-19-09810-t001:** Risk factors concerning the parent or caregiver, family situation, and child.

Parent or Caregiver	Family Situation	Child
Maltreatment experienced by parents in their childhood and prior trauma ^1–2^ Insufficient knowledge about child development ^3^ Difficulties in interpersonal relationships ^4,14^ Young age ^5^ No or low level of education ^6^ Experience of stress ^7^ Physical problems ^8–9,13–15^ Antisocial behavior and crime ^10^ Mental health problems ^11–15,28^ Insufficient understanding of the basic needs of the child ^13–15^	Single or divorced parent ^16^ Large family size; more than 3 children in the family ^17^ Low socioeconomic status ^18^ Very young children in the family ^19^ Experience of no or low social support ^20^ Intimate partner violence ^21^ Family violence and problems in close relationships ^21–22^	Problems with the child during the perinatal period ^23^ Excessively crying infant/child ^24^ A child with a disability or autism ^25^ A child with ADHD or difficult behavior ^26^ A child with learning problems ^27^

Superscripts numbers refers to the statement numbers of the final FNC in Section 3.3.

**Table 2 ijerph-19-09810-t002:** Comparison of the FNC family-related risk factors with risk factors identified in the WHO INSPIRE strategy, CDC, Scotland, and Finland.

FNC	WHO	CDC	Scotland	Finland
Maltreatment experienced by parents in their childhood and prior trauma	x	x	x	x
Insufficient knowledge about child development		x		
Difficulties in interpersonal relationships		x	x	
Young parent	x	x	x	
No or low level of education	x	x	x	
Experience of stress		x	x	x
Physical problems	x	x	x	x
Antisocial behavior and crime	x	x	x	
Mental health problems	x	x	x	x
Insufficient understanding of the basic needs of the child			x	
Single or divorced parent	x	x	x	x
Large family size: more than 3 children in the family		x		x
Low socioeconomic status		x	x	x
Very young children in the family	x	x	x	
Experience of no or low social support	x		x	x
Intimate partner violence	x	x	x	x
Family violence and problems in close relationships	x	x	x	x
A child with problems during the perinatal period				x
Excessively crying infant/child				
A child with a disability or autism	x	x	x	x
A child with ADHD or difficult behavior			x	x
A child with learning problems				
**Risk factors not included in the FNC**				
Sex	x			
Associating with delinquent peers	x			
Nonbiological parents		x		
Forced marriage	x			
Chaotic lifestyles of the parent, inadequate housing			x	
**Risk factors found in the FNC Feedback section**				
Lack of emotional bonding	x			
Poor parenting practices	x		x	

## Data Availability

The FNC trainer and professional manual is available through the ERICA webpages https://projects.tuni.fi/erica/ (accessed on 20 April 2022).

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
