# Peer review of "Family Needs Checklist: Development of a Mobile Application for Parents with Children to Assess the Risk for Child Maltreatment"

_ijerph, 2022, doi:10.3390/ijerph19169810_

Round 1

Reviewer 1 Report

Thank-you for offering me the opportunity to review your paper. This is an important topic and there is a great need for preventative measures to address risks of child maltreatment before it occurs. You have reviewed the evidence to support the development of your checklist application which will support practice in health and community settings. My key concerns are the clarity of your paper, and the methods utilised to conduct the systematic review. Some revision and refinement is needed to ensure your paper highlights the key messages of your work. More detailed comments are provided 

Introduction

Needs to be edited for clarity and sentence structure, a number of sentences do not have a period after the citation. Sentences should also not commence with an acronym.There are also a number of very long sentences that need to be refined.

The paragraph defining CM should come first, then follow on with the statistics presented in the first paragraph.

Overall the introduction is quite long and could do with some refinement to focus research on the key aims of the study 

Materials and Methods

You suggest that this was a systematic review update, have you published or someone else published a systematic review on this topic? If so it should be referenced.

'Another reason why we did not rely solely 215 on the previous research evidence was that not only should we answer, what the risk 216 factors are but also why they are risk factors to be able to build the feedback section for 217 the parents in the online application.' -- It is unclear what you mean when you suggest that understanding why they are risk factors to be able to build a feedback section in the online application. Please clarify

JBI Guidelines need to be referenced

Why did you limit your search to the time frame between 2015 and 2021?

'A systematic review of data- 209 bases was completed to retrieve systematic reviews, scoping reviews, and meta-analyses 210 focusing on risk factors of CM in families with children and risk assessment tools.' --You state that your systematic review examined risk assessment tools, however no findings are presented regarding risk assessment tools found in the literature? Please explain why this is the case

More information on inclusion and exclusion criteria of studies included in your systematic review is needed. 

A PICO statement would be helpful to understand your target literature.

What was the reason for choosing Finland and Scotland to validate the findings? (i.e., why were these countries specifically chosen over others?)

What method did you use to determine the quality of evidence in your systematic review ? This information should be presented.

What was your evidence extraction process? A table outlining each included paper author, year, country, participants, research methods, key findings would be helpful to visualise your systematic review findings.

Based on the information included in the methods, it is not clear to me that you have conducted a systematic review.

Discussion

The discussion is very long and includes a lot of detail about how the factors in the checklist relate to current guidelines. The discussion needs to first summarise key findings and the describe what the findings mean. The value of your study and it's implications a lost in the volume of detail in the discussion. Significant refinement is needed to highlight key messages, what need has your study addressed? What benefit does your tool have over others? What more needs to be done?

Author Response

Dear Reviewer 1. Please find attached file of my reply

Reviewer 2 Report

Thank you very much for the opportunity to read and review the paper entitled: Family Needs Checklist: Development of a mobile application for parents with children to assess the risk for child maltreatment,

The goal of the article is to develop an instrument (Family Needs Checklist) that will be used to assess the risk of Child Maltreatment with the potential to be used as an instrument in the prevention of maltreatment.

I liked the article and I think it can be publishable after a couple of minor additions to the present article. The authors did a fine study, although I think the article would benefit from a wider theoretical framework. This is especially relevant to the etiological factors involved in child maltreatment.

For example, the authors did not reference a seminal paper by Belsky:

·        Belsky, J. (1993). Etiology of child maltreatment: A developmental ecological analysis. Psychological bulletin, 114(3), 413.

I would also like the authors to pay attention to the ego-depletion theory and its role in child maltreatment. In relation to this I suggest the authors to consult a paper:

·        Henschel, S., de Bruin, M., & Möhler, E. (2014). Self-control and child abuse potential in mothers with an abuse history and their preschool children. Journal of Child and Family Studies, 23(5), 824-836.

I would also like the authors to tell us more about the other strategies used for the prevention of child maltreatment, such as parental education.

At the end, I wish the authors all the best in their future efforts and hope the Family Needs Checklist will contribute to the reduction of child maltreatment.

Author Response

Dear Reviewer 2. Please find attached file of my reply

Reviewer 3 Report

Thank you for the opportunity to review this manuscript. The topic and idea are important and interesting, especially to the goal of eliminating child maltreatment. Please see my overall and specific comments below. 

Overall, I suggest rewriting the manuscript to clearly indicate the purpose and intended users of the checklist. It seems as though the purpose and primary users are unclear, this may be the way that I'm reading the manuscript but I was not sure about these areas. 

The manuscript does not discuss that the risk factors for CM are only for identified CM and that some risk factors are static (e.g., experienced maltreatment as a child). So, if a professional (e.g., doctor) uses the checklist and reports that their patient (e.g., a caregiver) was maltreated as a child, what would be the protocol? A discussion about reported risk factors and surveillance would promote some nuance into the discussion, as CM is complex, its prevention, therefore, is complex as well. 

Overall, the manuscript needs to be thoroughly edited for grammar and concision. Some edits and specific comments are below.

Page 1

Line 45 – missing period. Missing The at the beginning of sentence

Page 2

Line 46 – safety not safe lives

Line 71 & 72 – sentences need to be edited

Line 84-86 – reword

Line 95-96 – reword

Page 4

Line 157 – who is the “user”? What does “improving user outcomes” mean if the user is the professional? Isn’t the goal to improve family outcomes?

Page 6

Line 273 – the total number of meta-analyses and systematic reviews does not seem to add up properly

Page 9

Line 379-386 – seems to be the first time that the purpose and users of the checklist are clearly stated. Do they use this with child welfare/protection professionals?

Page 10

Line 406 – indicates how parents should use checklist – this should be identified much earlier in the manuscript

Page 12

Line 514 – reword

Line 530 – remove quotation mark

Page 13

Line 594 – this fulsome discussion of how the checklist may be used should be discussed earlier in the manuscript

Author Response

Dear Reviewer 3. Please find attached file of my reply
